# A Bingham Plastic Fluid Solver for Turbulent Flow of Dense Muddy Sediment Mixtures

Ian Adams [1,2,*], Julian Simeonov [2], Samuel Bateman [2] and Nathan Keane [3]

1 National Research Council Research Associateship Program, Washington, DC 20001, USA
2 Ocean Sciences Division U.S. Naval Research Laboratory, Hancock County, MS 39529, USA; julian.simeonov@nrlssc.navy.mil (J.S.); sam.bateman@nrlssc.navy.mil (S.B.)
3 College of Earth Ocean and Atmospheric Sciences, Oregon State University, Corvallis, OR 97331, USA; keanen@oregonstate.edu
* Correspondence: ian.adams.ctr@nrlssc.navy.mil

**Abstract:** We have developed and tested a numerical model for turbulence resolving simulations of dense mud–water mixtures in oscillatory bottom boundary layers, based on a low Stokes number formulation of the two-phase equations. The resulting non-Boussinesq equation for the fluid momentum is coupled to a transport equation for the mud volumetric concentration, giving rise to a volume-averaged fluid velocity that is non-solenoidal, and the model was implemented as a new compressible flow solver. An oscillating pressure gradient force was implemented in the correction step of the standard semi-implicit method for pressure linked equations (SIMPLE), for consistency with the treatment of other volume forces (e.g., gravity). The flow solver was further coupled to a new library for Bingham plastic materials, in order to model the rheological properties of dense mud mixtures using empirically determined concentration-dependent yield stress and viscosity. We present three direct numerical simulation tests to validate the new MudMixtureFoam solver against previous numerical solutions and experimental data. The first considered steady flow of Bingham plastic fluid with uniform concentration around a sphere, with Bingham numbers ranging from 1 to 100 and Reynolds numbers ranging from 0.1 to 100. The second considered the development of turbulence in oscillatory bottom boundary layer flow, and showed the formation of an intermittently turbulent layer with peak velocity perturbations exceeding 10 percent of the freestream flow velocity and occurring at a distance from the bottom comparable to the Stokes boundary layer thickness. The third considered the effects of density stratification due to resuspended sediment on turbulence in oscillatory bottom boundary layer flow with a bulk Richardson number of $1 \times 10^{-4}$ and a Stokes–Reynolds number of 1000, and showed the formation of a lutocline between 20 and 40 Stokes boundary layer depths. In all cases, the new solver produced excellent agreement with the previous results.

**Keywords:** Bingham plastic; OpenFOAM; sediment; viscoplastic; rheology; turbulence

## 1. Introduction

Resuspension and transport of sediment are the defining processes in the continual evolution of aqueous topography in shallow coastal environments. Morphological changes in ocean, estuary, and river environments impact shoreline change, bottom topography, nearshore navigation in a changing environment, acoustic communication and observation, and seabed interaction with man-made structures. As such, understanding the mechanisms of resuspension and transport is of critical importance to the development of an understanding of these coastal environments. In the bottom boundary layer, resuspension and fluidization of bed sediments are intimately coupled to the turbulent shear stresses generated at the interface between the fluid and the bed [1,2].

The focus of this paper and the associated solver is the shallow coastal environments characterized by large concentrations of cohesive dense muddy sediments (e.g., in the

Gulf of Mexico). Of particular importance in the shallow coastal environment is the dominant influence of ocean surface waves [3,4], which generate the shear stresses needed to resuspend sediment from the bed, as well as to transport the sediment bedload. Wave-induced sediment resuspension can support gravity flows that drive a large portion of cross-shelf fine sediment transport [5]. Under an oscillating wave, turbulence is limited to a thin boundary layer just above the bed. The strong shear disturbs the bed sediment, fluidizing a large concentration of muddy sediment in this layer. Transport of high concentration sediment mixtures in the near bed layer is a large contributor to the evolution of the morphology in shallow coastal regions [6,7].

Dense muddy mixtures of cohesive sediments exhibit a non-Newtonian behavior that can be approximated as a Bingham plastic [8,9]. Comparisons between the Newtonian model and the Bingham model have shown that the dynamics of dense muddy mixtures are significantly changed by the presence of a yield stress [8], which clearly makes the task of properly modeling this yield stress important. Models for the yield stress of the mixture have been proposed based on empirical studies of wave flow interactions with cohesive sediments, where relationships were determined using yield stress estimates as a function of sediment concentrations [10,11].

We presently do not understand how the non-Newtonian concentration-dependent rheology affects the structure of the turbulence in the boundary layer. It has been shown that within the bottom boundary layer, the viscoplastic rheological properties of the dense muddy mixture reduce the turbulent kinetic energy by increasing dissipation [12]. Likewise, the stable stratification resulting from wave-induced resuspension of bottom sediments has been seen to both damp turbulent kinetic energy and reduce the bed shear stress [13]. The interaction between this damping and the turbulent resuspension and transport of muddy sediment is not yet fully understood, but recent research has indicated a strong relationship between small scale turbulence in the bottom boundary layer and the movement of fine muddy sediment [14,15]. Two-phase modeling has been used to show that the boundary layer thickness of fine sediment suspension is increased by the complex interplay between flow instabilities, stress terms, and sediment-induced turbulence attenuation [16]. It is worthwhile to mention the processes of erosion within the context of the rheological behavior of the dense muddy mixtures. Typically, the flux of sediment into the water column from the bed is modeled as the product of an erosion rate, a probability factor, and a function of the shear stress exceeding the critical shear stress [17], with some modifications required in different scenarios, such as cases of freshly deposited muddy beds [18]. In contrast, the goal for the new solver considered here is to explicitly resolve the erosion of sediment from the dense lower layers into the water column under an oscillating flow. Depending on the Reynolds number of the oscillating conditions, previous direct numerical simulations have found four different flow regimes: the fully laminar regime, disturbed-laminar regime, intermittently turbulent regime, and fully turbulent regime [19]. Of particular interest here is the regime of intermittent turbulence, which has been investigated with numerical simulation [19] and is later used as a benchmark for our model. The aforementioned simulations did not consider the non-Newtonian rheology of muddy sediments and its dependence on the sediment concentration within the fluid.

Previous numerical investigations of the turbulent flow of mud mixtures made a Boussinesq approximation and assumed that the fluid flow was incompressible, which is generally valid when the concentration of sediment is sufficiently small [15,20,21]. However, this approach is not adequate when explicitly modeling the erosion process from large concentrations near the bottom. Other models such as SedFoam 2.0 [22] are non-Boussinesq, have options for different sediment rheologies, and are based on the full two-phase momentum equations. Here, we focus on the simpler approach based on the low Stokes number simplification of the two phase equations, where the dynamics are governed by the fluid momentum equation. Our non-Boussinesq model is coupled with Bingham plastic rheology, where viscoplasticity is represented by a dynamic viscosity and yield stress that are specifically concentration-dependent. This is done by utilizing functions of dynamic

viscosity and yield stress that obey empirically determined dependencies on sediment concentration. We should mention that the existing viscoplastic rheology in OpenFOAM standard libraries [23] uses a concentration-independent yield stress and viscosity and is unsuited for modeling dense muddy mixtures.

This paper is organized into several sections. First, we will address the physics foundation of the solver we have developed. This will include a discussion of the development of the necessary two-phase conservation equations for fluid and sediment used within our new solver, the development of the momentum equations that govern the dynamics of the mud–water mixture within the mudMixtureFoam solver, and the treatment of the viscoplasticity terms for rheological considerations of dense mud mixtures. Second, the implementation of the mathematical theory into OpenFOAM will be discussed. Third, tests will be reported that demonstrate the capability and strength of the solver to accurately model the rheological properties of dense Bingham fluids compared to simulations of Bingham flows over a sphere [24], as well as the generation of turbulence within the bottom boundary layer under both single-phase and two-phase oscillatory flow.

## 2. Mathematical Formulation

As has been the convention in previous work [15,20], we assume that the fine sediment particles of low Stokes number move with the fluid, having a velocity $u_s$ that is identical to the fluid velocity $u_f$, save for in the vertical, where the downward sediment velocity exceeds the fluid velocity by an extent equal to the settling velocity of the particle, shown by

$$u_i^s = u_i^f - W_s \delta_{i3} \tag{1}$$

where the settling velocity, $W_S$, includes a hindered settling effect, which decreases the settling velocity as the sediment becomes more dense. In order to conserve the total sediment concentration within the domain, the settling velocity must approach zero at the bottom boundary, such that there is no export of sediment across the bottom boundary. This was accomplished with use of a ramp-down function, causing the total amount of sediment in the domain to be fixed. We neglect any elastic behavior that occurs in the thin layer near the bottom boundary. In suspensions of cohesive sediments, the settling velocity of the particles is dependent on the way in which they flocculate into aggregates or flocs. Many attempts have been made to model the complicated interaction between floc aggregation/breakup and the setting velocity, ranging from simple empirical relationships based on instantaneous concentration and turbulent dissipation rate, to kinetic equations describing the aggregation and breakup of flocs [25], to population equation balance and size class modeling [26,27]. Modeling these flocculation effects is computationally demanding, and it is more efficient to select a representative settling velocity for the sediment. In this model, we consider concentration-dependent settling velocities representative of sediment in dense muddy mixtures given by

$$W_s = W_{s,0}\left(1 - \frac{\phi}{\phi_{max}}\right)^{m_w} \tag{2}$$

where $W_{s,0}$ is the reference settling velocity of the sediment, $\phi_{max}$ is the maximum allowed concentration by volume, and $m_w$ is an empirically determined constant [28]. Two-phase conservation equations for the mass of the fluid and sediment phases are

$$\frac{\partial(1-\phi)}{\partial t} + \frac{\partial(1-\phi)u_i^f}{\partial x_i} = -\frac{\nu_t \partial^2 \phi}{Sc \partial x_i^2} \tag{3}$$

$$\frac{\partial\phi}{\partial t} + \frac{\partial\phi u_i^s}{\partial x_i} = \frac{\nu_t \partial^2 \phi}{Sc \partial x_i^2} \tag{4}$$

where the terms on the left-hand side of both equations are the change with respect to time and the divergence of the velocity, respectively. In addition, when the flows are not entirely resolved, it becomes necessary to model the unresolved terms with a diffusion term, which

is a function of the turbulent viscosity and the Schmidt number. The viscosity, $\nu_t$, is a sum of the turbulent and molecular viscosities, and the Schmidt number is the dimensionless ratio of viscous diffusion to molecular diffusion (taken here to be $Sc = 0.5$). This diffusion term is typically introduced to increase the numerical stability of simulations.

To determine the momentum of a suspension of fluidized particles within a fluid, it is necessary to obtain averaged values of the velocity of the fluid, velocities of the particulates, and fluid pressure. This method simplifies the approach, allowing us to apply the Navier–Stokes equations to representative regions of fluid. Local spatial averages taken for the fluid and solid phases require the definition of the portion of the volume occupied by the fluid, $\epsilon$ (also referred to as the void fraction), and the portion of the volume occupied by the solid, $\phi$. Reynolds averaging over the momentum equation produces temporally averaged terms, where quantities are separated into their mean and fluctuating components.

The basis of our model equations rests on the volume-averaged two-phase fluid momentum equations derived by Jackson [29]. As the fine sediment follows the fluid flow, except with respect to the settling velocity, it is unnecessary to fully solve both the fluid and particle momentum phase equations. Instead, the known velocity relationship between the fluid and fine, low Stokes number clay particles results in a simplification, such that the dynamics of the mixture is governed by the fluid phase momentum equation alone. The fluid phase momentum equation is given as

$$\rho_f \epsilon \frac{D\mathbf{u}^f}{Dt} = \nabla \cdot \mathbf{S}_f - nf + \rho_f \epsilon \mathbf{g} \tag{5}$$

where the term on the left hand side represents the material derivative of the fluid, $\frac{D}{Dt} = \frac{\partial}{\partial t} + \mathbf{u} \cdot \nabla$. $\rho_f$ the fluid density, $\mathbf{S}_f$ the fluid averaged effective stress tensor, and $f$ represents the average value of the force exerted by the fluid on a particle. Using Jackson's definition of the stress tensor

$$\nabla \cdot \mathbf{S}_f = -\nabla p + \nabla \cdot \nu_{eff} \nabla \mathbf{u} \tag{6}$$

we can recast the fluid momentum equation in terms of the pressure and the dissipation terms as follows:

$$\rho_f \epsilon \frac{D\mathbf{u}^f}{Dt} = -\nabla p + \nabla \cdot \nu_{eff} \nabla \mathbf{u}^f - nf + \rho_f \epsilon \mathbf{g} \tag{7}$$

where p is the fluid pressure and $\nu_{eff}$ is an effective fluid viscosity that combines $\nu_t$ with a mud mixture viscosity approximating a Bingham plastic. Momentum exchange terms in the fluid equation are disregarded as they are dependent on small settling velocities and only act in the vertical direction.

The key consideration in the present model is the inclusion of a cohesive sediment rheology suitable for dense muddy sediment. Rheometer measurements have suggested that muddy mixtures closely follow a Bingham plastic rheology [9], where the effective viscosity is increasingly large under conditions of low shear flow. To account for the Bingham flow viscoplactic contributions, we model a regularized effective viscosity that approximates the stress of a Bingham plastic. In the current low Stokes number regime, the entire mixture viscosity stress is accounted for in the fluid momentum balance as given by

$$\nu_{mud} = \frac{1}{\rho_w}\left(\mu_{mud} + \frac{\tau_y}{\dot{\gamma}+\gamma_{min}}[1 - exp(-m(\dot{\gamma}+\dot{\gamma}_{min}))]\right) \tag{8}$$

where $\mu_{mud}$ is the dynamic viscosity of the muddy mixture, $\tau_y$ is the yield stress of the mixture, and $\dot{\gamma}$ is the shear rate. The constant $m$ is a regularization parameter chosen to ensure that the rheology asymptotically approaches Bingham plastic behavior for cases where $m\dot{\gamma} >> 1$, while for cases where $m\dot{\gamma} << 1$, the rheology asymptotically approaches high-viscosity Newtonian fluid behavior. Empirically determined coefficients for the yield stress, dynamic viscosity, and mixture density were obtained based on data from Komatina [11].

### 3. Numerical Implementation with OpenFOAM

Here we describe a new numerical solver, mudMixtureFoam, that uses the computational fluid dynamics framework of OpenFOAM [30] to solve Equations (1)–(8). To solve for the non-Bousinesq compressible fluid flow in Equations (3) and (7), we modify the buoyantPimpleFoam solver for compressible, transient flow, as described in detail below.

We are interested in turbulent flows forced by an oscillating pressure gradient. In OpenFOAM solvers, it is possible to include pressure forcing in the velocity equation of the momentum prediction step of the pressure–velocity coupling. However, when the flow is transient, this approach can result in numerical instabilities, and the alternative approach adopted in OpenFOAM is to include the forcing term in the correction step of the standard semi-implicit method for pressure linked equations (SIMPLE) [31]. Since the aim of this model is to simulate turbulent, transient flows of a fully two-phase muddy mixture, we use the later approach to implement an oscillating pressure gradient force. The flow solver was further coupled to a new library for Bingham plastic materials to incorporate the effects of concentration-dependent viscoplastic flow viscosity and yield stress (Equation (8)).

Following the general Rhie and Chow procedure [32] of the finite-volume method, the semi-discretized form of the fluid-phase momentum equation is

$$a_P U = H(U) - \nabla p_{rgh} + \nabla \cdot \nu_{eff} \nabla u + F \tag{9}$$

where $a_P$ is the diagonal term of the matrix resulting from the discretization of the convection term, and H(U) is the off-diagonal term of velocity, both functions of the velocity field and concentration. F is a body force term, and $p_{rgh}$ is the alternate pressure (i.e., the dynamic pressure) defined by the removal of the hydrostatic pressure $p_{rgh} = p - \rho(g \cdot h)$, where h is some arbitrary height.

The SIMPLE method constructs a convection matrix $a_P$ and $H(U)$ and makes a prediction estimate of momentum from Equation (9) using the pressure from the previous time step. It then estimates the pressure field from the following Poisson equation

$$\nabla \cdot \left[ (1 - \phi) \frac{H(U)}{a_P} - (1 - \phi) \frac{\nabla p_{rgh}}{a_P} + (1 - \phi) \frac{F}{a_P} \right] + \frac{\partial \epsilon}{\partial t} = -\frac{\nu_t \partial^2 \phi}{Sc \partial x_i^2} \tag{10}$$

obtained by substituting the velocity from Equation (9) into the equation for void fraction conservation (Equation (3)). The final SIMPLE step is the correction step, where velocity is estimated with the updated pressure. The steps of estimating the pressure and correcting the velocity field in the correction step are also known as the PISO method [33], which can be iterated within each SIMPLE iteration.

The result is a novel numerical solver within the OpenFOAM framework that can be applied to dense muddy mixtures of fine suspended sediment. The treatment of the time derivative of the void concentration is a straightforward implementation from buoyantPimplefoam (which the mudMixtureFoam solver was adapted from). The novel aspect is that the body forcing term includes not only gravity but also the oscillatory pressure gradient forcing. High accuracy discretization schemes were chosen for the convection–diffusion terms and the temporal terms (central differencing and Crank–Nicholson, respectively), and the pressure gradient force included in the correction step of the velocity–pressure coupling loop applies a user defined oscillating flow to the mixture.

### 4. Results and Discussion

*4.1. Viscoplastic Flow around a Sphere*

We first validate the ability of the mudMixtureFoam solver to simulate the rheological properties of a Bingham fluid passing over a stationary spherical particle. This is carried out by comparing the result of mudMixtureFoam to the work of Gavrilov et al. [24], who modeled a Bingham plastic flowing past a sphere with a different finite-volume solver. The properties that we have chosen to compare are the surface drag over the sphere and the

viscous stress distribution in the Bingham fluid in the vicinity of the sphere surface, two properties of the fluid that depend heavily on the rheology.

Following the methodology of Gavrilov et al. [24], we utilize radial symmetry to minimize the computational domain necessary to model. We construct a 3D domain of extent $Lx \times Ly \times Lz$ with a solid spherical surface centered at $(Lx/2, 0, 0)$. This results in a quarter of the solid sphere extending into our domain (Figure 1). Using radial symmetry, we can extrapolate that the unmodeled volumes are symmetrical reflections across the y = 0 plane and the z = 0 plane centered at $(Lx/2, 0, 0)$.

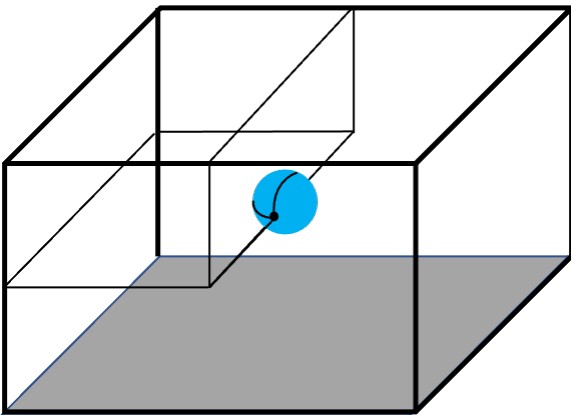

**Figure 1.** Diagram of the modeled area respective to the sphere. The upper left-hand quadrant is the model domain, with assumed symmetry conditions on the interfaces between quadrants.

The domain size is set to cover a distance 50 radii from the sphere center. A circumferential resolution of 72 cells is defined at the surface of the sphere (Figure 2), with this high-resolution area extending to a distance five radii from the sphere center. A constant fluid flow aligned along the x-axis is used to initialize the simulation and is also imposed as an inlet boundary condition with magnitudes indicated in Table 1. The surface of the sphere is set to be a no-slip wall boundary, with the other domain boundaries being assigned the symmetry boundary condition. The forward Euler scheme was used for time integration of the governing equations. The simulated flow field quickly converged to the final steady state for the low Reynolds number considered in the present tests.

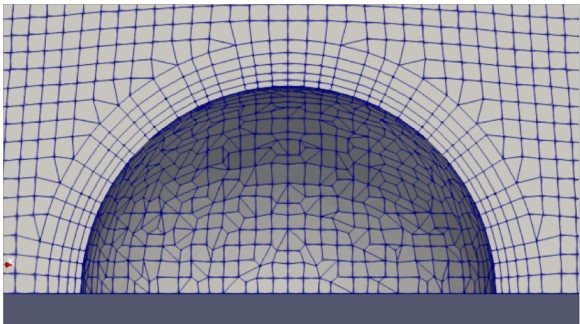

**Figure 2.** Near-surface mesh around the sphere. Five surface layers are implemented with an expansion ratio of 1.2.

**Table 1.** Bingham Fluid Parameters.

| Case | Bn | Re | V (mm/s) | D (mm) |
|------|-----|------|----------|--------|
| A | 1 | 0.1 | 2 | 0.172 |
| B | 100 | 1.0 | 0.643 | 5.6 |
| C | 1 | 100 | 64.3 | 5.6 |

Drag and viscosity are intimately related to the system rheology. As fluid flow encounters the solid sphere and is deflected around the sphere surface, the no-slip wall boundary creates a shear stress around the sphere, which decreases the viscosity of the Bingham fluid in the region where the yield stress is exceeded. Using his simulations Gavrilov [24] obtained the following empirical drag force on the sphere

$$Fd = 3\pi\mu_f DV(1 + 0.15Re^{0.687} + 1.79Bn^{1/2} + 1.09Bn) \tag{11}$$

as a function of the Reynolds number (*Re*) of the fluid

$$Re = \frac{\rho_m DV}{\mu_m}, \tag{12}$$

and the Bingham number (*Bn*) of the fluid

$$Bn = \frac{\tau_y D}{\mu_m V}, \tag{13}$$

where $D$ is the sphere diameter, V is the velocity of the fluid, $\rho_m$ is the fluid density, $\mu_m$ is the dynamic viscosity, and $\tau_y$ is the yield stress; the last three properties would be functions of the sediment concentration in the case of mud suspensions.

We considered three unique cases of steady Bingham fluid flow over a range of $1 < Bn < 100$ and Reynolds numbers from $0.1 < Re < 100$ (shown in Table 1). Sphere diameter and fluid velocity were varied to obtain a range of Reynolds and Bingham numbers, while the yield stress, density, and viscosity were held constant.

Drag-induced shear causes stress near the sphere surface, which in turn results in a lowered effective viscosity of the Bingham fluid near the sphere surface. As the Bingham number decreases, or Reynolds number increases, we expect that the flow and shear effects would more strongly dominate, resulting in a larger area of lowered viscosity around the sphere (as observed in Figures 3–5). These figures represent contour cross-sections of the fluid viscosity through the center of the rigid sphere. Figures 3 and 5 correspond directly to cases A and C, respectively, as examined by Gavrilov et al. [24]. Case B (Figure 4) was undertaken to demonstrate the ability of mudMixtureFoam to handle fluids with very large Bingham numbers.

The contour lines correspond to the following range of normalized values of effective viscosity: $1 < \nu/k(1 + Bn) < 10$. In case A, the region of lowered viscosity forms a symmetrical bubble around the rigid sphere, with a smaller ring of higher viscosity halfway around the sphere. As the Bingham number increases, the region of lowered viscosity deforms somewhat, extending outwards in the upstream and downstream directions, and a distinct divot forms at the top. Presumably, this can be attributed to the effects of increased yield stress causing the fluid to resist shear near the sphere surface. Predictably, this causes the region of lowered viscosity to have a more limited extension in the cross-stream direction away from the sphere. When the Reynolds number is dramatically increased, as in case C, the region of lowered viscosity can be seen to lengthen quite dramatically downstream. This is expected, as at high Reynolds numbers, the inertial forces are strong enough to overcome the viscous forces, and the fluid stress exceeds the yield stress of the Bingham fluid, both near the sphere surface and downstream of the rigid sphere, creating a wake of lowered viscosity.

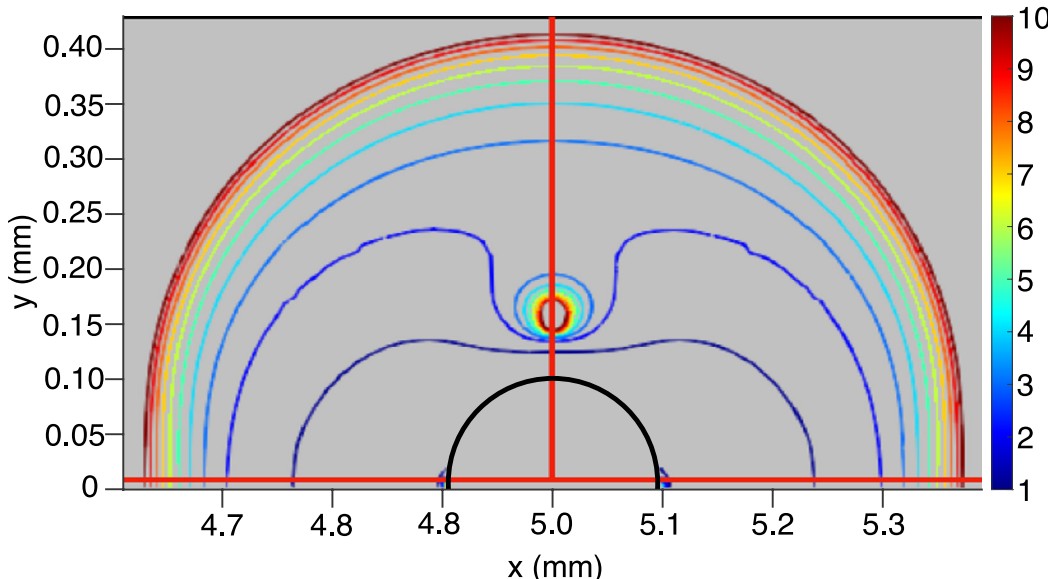

**Figure 3.** A vertical cross-section of the viscosity bubble around the solid sphere (black line) is shown for Case A with Re = 0.1, Bn = 1. Colored lines correspond to the viscosity, nondimensionalized by $k(1 + Bn)$. The contours are integer values in the range 1–10.

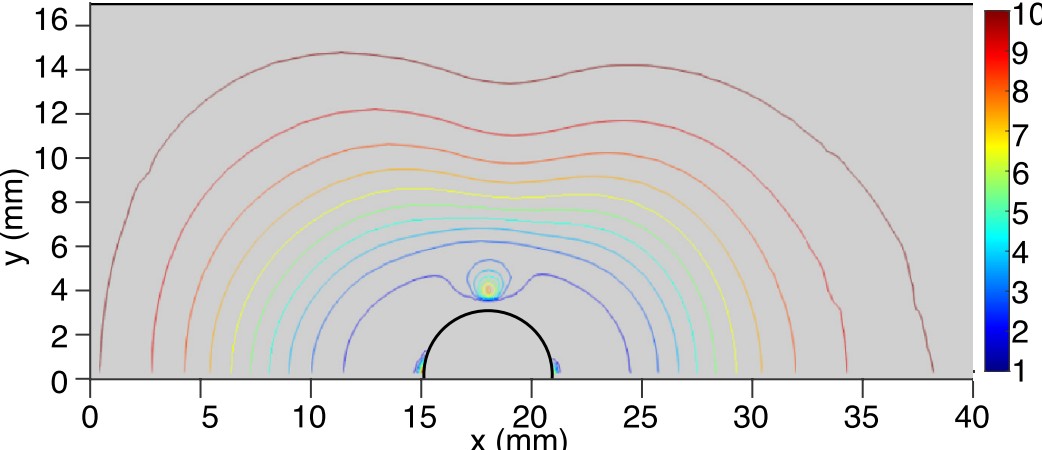

**Figure 4.** A vertical cross-section of the viscosity bubble around the solid sphere (black line) is shown for Case B with Re = 1, Bn = 100. Colored lines correspond to the viscosity, nondimensionalized by $k(1 + Bn)$. The contours are integer values in the range 1–10.

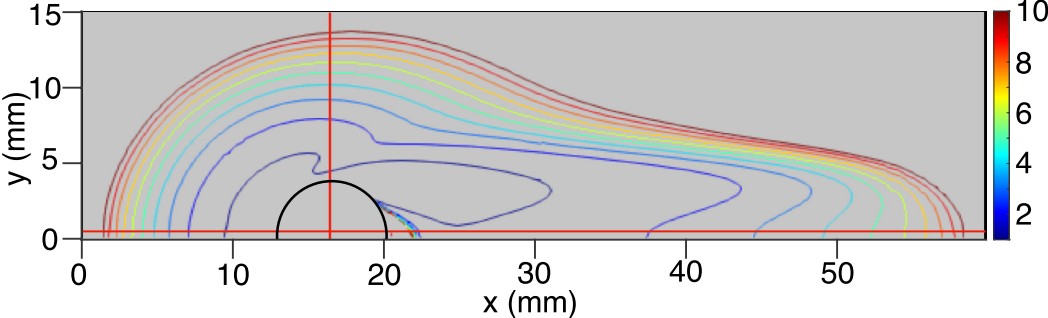

**Figure 5.** A vertical cross-section of the viscosity bubble around the solid sphere (black line) is shown for Case C with Re = 100, Bn = 1. Colored lines correspond to the viscosity, nondimensionalized by $k(1 + Bn)$. The contours are integer values in the range 1–10.

Figures 6 and 7 display the vertical and horizontal cross-sections of viscosity taken from cases A and C, compared with the contour profiles of data digitized from the corresponding Gavrilov results.

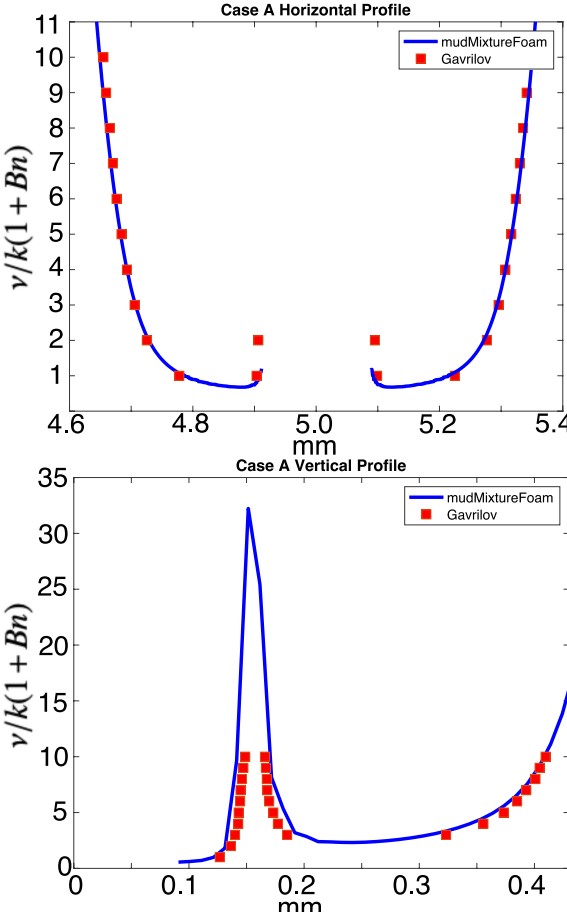

**Figure 6.** Comparison of case A horizontal and vertical profiles of nondimensionalized viscosity through the sphere with the corresponding profiles in Gavrilov.

The profiles of viscosity through the sphere determined by the mudMixtureFoam solver show a good match with the simulation results from Gavrilov et al. [24]. A quantitative comparison of the viscosity profiles to Gavrilov et al. [24] was made by calculation of the root-mean-square error (RMSE), normalized using the viscosity range. Case A shows the best fit of the model with the established simulation, with a normalized RMSE of 0.0135 in the vertical and 0.0011 in the horizontal. Case C shows an excellent match in the vertical, with a normalized RMSE of 0.000237. The horizontal profile of Case C shows a good match on the upstream side (normalized RMSE = 0.000379), but on the downstream there is a clear under-prediction of the extent of the region of lowered viscosity. This resulted in a normalized RSME value of 0.195. This is a notable divergence from the Gavrilov simulation results, but was likely a result of the limited extent of downstream refinements used within the OpenFOAM domain.

Successful modeling of drag force by mudMixtureFoam would produce values for drag force over the sphere surface consistent with the theoretical drag force obtained from Equation (11). Figure 8 demonstrates that the modeled value of the drag converged toward the empirical value over the last several hundred model iterations of the test case runs.

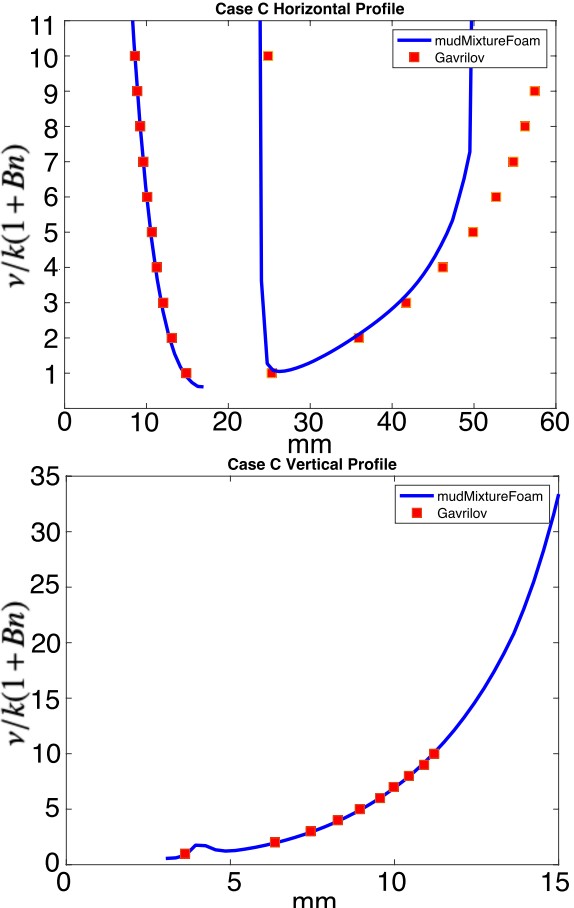

**Figure 7.** Comparison of case C horizontal and vertical profiles of nondimensionalized viscosity through the sphere with the corresponding profiles in Gavrilov.

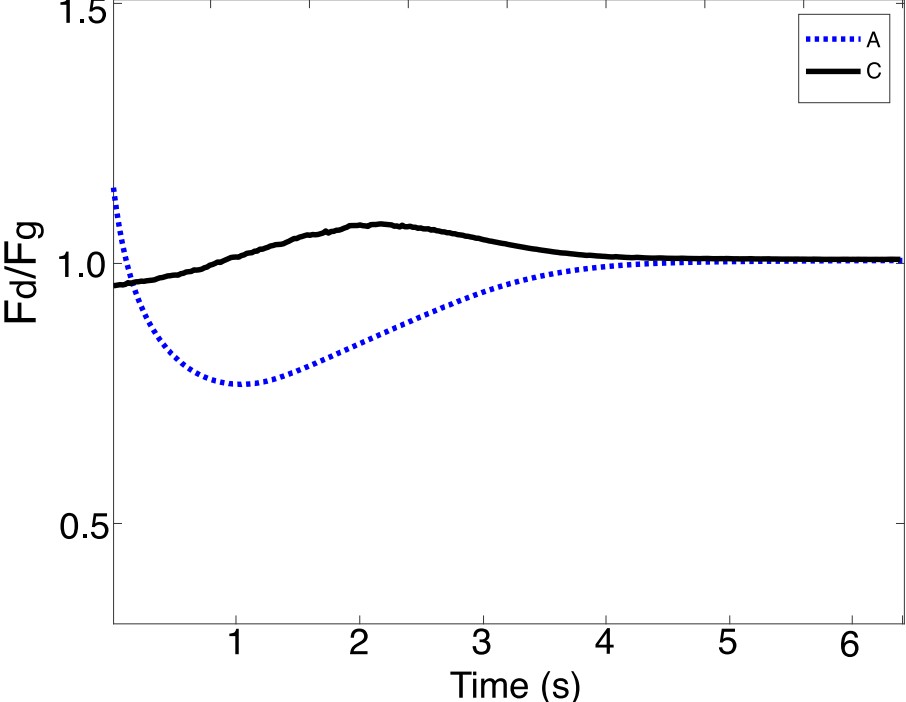

**Figure 8.** Ratio of $F_d$, the modeled drag coefficient over the sphere surface, to $F_g$, the expected drag force calculated using Equation (11).

Taken together, the examination of drag and viscosity clearly demonstrate that the new mudMixtureFoam solver is able to reliably model the rheology of a Bingham plastic fluid, making it a suitable method for modeling areas with dense muddy sediment. With this validation of the rheological component of the model, we turn to the examination of the ability of mudMixtureFoam to appropriately handle the generation of turbulence in the bottom boundary layer.

### 4.2. Turbulence in Single-Phase Oscillatory Boundary Layer

The ultimate goal of the mudMixtureFoam solver is to model turbulent resuspension of sediments and their transport in the bottom boundary layer. There is a large body of simulation work concerning bottom boundary later turbulence under oscillating flow conditions [15,19,20,34,35], and we have selected here the work of Costamagna et al. [36] to serve as a benchmark for our model. There are two facets to the motivation for this section. The first is to show that we can replicate the growth and behavior of intermittent turbulence in the bottom boundary layer, and the second is to test the stability of our solver for non-solenoidal flow under conditions of non-uniform sediment concentration.

We consider a 3D domain (a small box near the bottom boundary) analogous to that used by Costamagna. The domain extent is a multiple of the Stokes boundary length scale

$$\Delta = \sqrt{\frac{2\nu_f}{\omega}}, \tag{14}$$

where $\omega$ is the angular frequency of the oscillatory flow. Costamagna et al. [36] tested the domain size necessary to produce turbulent effects, and a domain with dimensions $(50.27\Delta \times 25.14\Delta \times 25.14\Delta)$ was sufficient to produce a good match when compared to experimental observation [37]. While subsequent studies have expanded this domain to produce more targeted turbulence effects, such as tracking turbulent spots within the flow [34], for a simple comparison we deemed the smaller domain sufficient. Periodic boundary conditions were selected on the horizontal boundaries, a no-slip wall condition was forced at the bottom boundary, and a slip boundary condition was selected for the upper boundary.

Flow through the domain was forced by the oscillating pressure gradient term

$$\frac{\partial P}{\partial x} = \rho_f \omega U_0 \cos(\omega t) \tag{15}$$

where $\omega$ is the angular frequency of the oscillating wave, given by $\omega = 2\pi/T$ and where T is the period of the wave. This term corresponds to a free stream wave with velocity $U = U_0 \sin(\omega t)$. A period of five seconds was selected, resulting in an angular frequency $\omega = 1.257 \text{ s}^{-1}$ and Stokes boundary layer thickness $\Delta = 1.26$ mm. In oscillatory flow, the flow regime of laminar, disturbed laminar, intermittent turbulent, or fully turbulent is typically described using the Stokes Reynolds number [38,39]

$$Re_\Delta = \frac{U_0 \Delta}{\nu_f} \tag{16}$$

where the intermittently turbulent regime exists in the range of $400 < Re_\Delta < 1200$. To keep consistency with with energetic shelf wave conditions, and to produce Stokes–Reynolds numbers of the same level as those used by Costamagna et al. [36], a $U_0$ of 0.752 m/s was selected. The natural transition to turbulence from laminar flow can be triggered by initially small variations induced by perturbations, such as wall waviness or vibration. These cause small variations from the laminar regime, which grow and cause the breakdown of laminar flow into turbulence. An infinitesimal waviness comprised of a superposition of sinusoidal waves was applied to the bottom boundary of our domain

$$x_3 = 0.005\Delta \sum_{n=1}^{N} a_n \cos(\alpha_n x_1 + \gamma_n x_2 + \phi_n) \tag{17}$$

where $x_1$ and $x_2$ represent the nondimensionalized $x$ and $y$ positions, respectively, and $x_3$ is the vertical position of the bottom boundary. This expression is characterized by the wavenumbers $\alpha_n$ in the x dimension and $\gamma_n$ in the y dimension. The parameters for waviness used in this study are given in Table 2.

**Table 2.** Waviness parameters.

| $a_1$ | $a_2$ | $\alpha_1$ | $\alpha_2$ | $\gamma_1$ | $\gamma_2$ | $\phi_1$ | $\phi_2$ |
|-------|-------|-----------|-----------|-----------|-----------|---------|---------|
| 1 | 0.1 | 0.5 | 0 | 0 | 1 | 0 | 0 |

Taking into account all the factors above, we chose our domain to be of size 6.33 cm $\times$ 3.17 cm $\times$ 3.17 cm, with 192 $\times$ 64 $\times$ 96 grid points. The infinitesimal bottom boundary waviness described by Equation (17) was introduced by calculating a spline fit through a series of points.

We studied two specific case runs. The first had a zero sediment concentration throughout the domain, for a direct comparison of our model results to those of Costamagna et al. [36]. The second case introduced a very small sediment concentration, initialized using the exponential profile:

$$c = 0.00013 * exp(-27.7z) \tag{18}$$

to demonstrate that the solver can perform calculations with concentration-dependent terms. In the intermittently turbulent regime, the turbulent kinetic energy (TKE) is characterized by an oscillation, where turbulence grows during the early part of the deceleration phase and diminishes during the early phases of accelerating flow. Figure 9 shows the free-stream mean velocity, the oscillating pressure gradient forcing term, and the domain-averaged turbulent kinetic energy. The first simulated cycle served as a ramp-up of the model, while phase averages were taken over the second and third cycles. We begin with a quiescent fluid with an initially large pressure gradient, which follows a sinusoidal behavior through one cycle. This results in a velocity oscillation which has the expected $\pi/2$ phase shift. For our five-second cycle, this means we expect maximum velocity magnitudes at $\pi/2$ and $3\pi/2$ of each cycle, which can clearly be seen in Figure 9. Turbulent kinetic energy follows the velocity oscillations, reaching maximum values at the early phases of decelerating cross-flow.

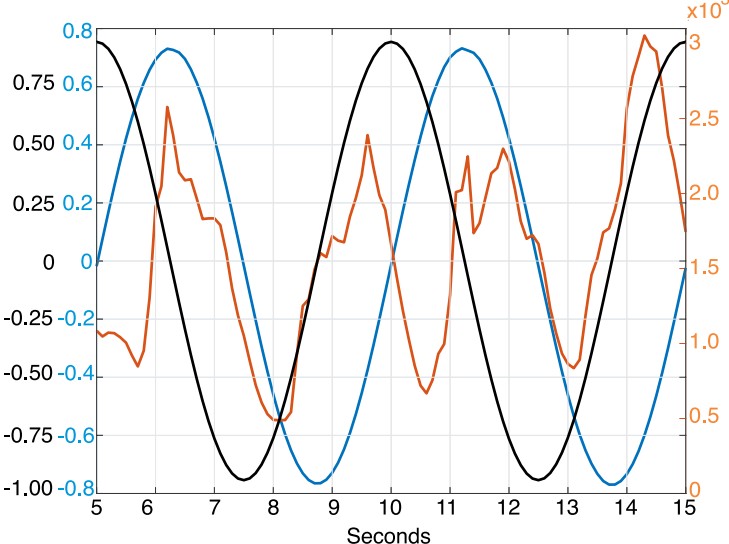

**Figure 9.** Free-stream velocity in m/s (blue), pressure gradient of $\omega \cos(\omega t)$ (black), and turbulent kinetic energy in m$^2$/s$^2$ (red).

Figure 10 shows a comparison between both mudMixtureFoam solver case runs and the digitized results of Costamagna et al. [36]. The streamwise velocity fluctuations normalized by $U_0$ decreased to values lower than 0.01 above 20 mm from the bottom. At $\pi$, where the oscillating flow is zero and the acceleration is negative, the maximum fluctuations occur between 5 mm and 10 mm, reaching values of up to 0.05 with regularity. At $1.24\,\pi$, when the oscillating flow is oriented in the negative x direction, peak fluctuations occur around 1 mm from the bottom (comparable to $\Delta$), reaching values of just under 0.09. At $1.64\pi$ under conditions of negative flow and positive acceleration, the profile of streamwise fluctuations is seen to increase in the region below 15 mm, reaching values up to 0.12 near the bottom. Normalized vertical velocity fluctuations have similar behaviors, where at $\pi$, the profile can be seen to have a large increase in values under 25 mm, with the largest values occurring between 15 mm and 5 mm, reaching magnitudes of just under 0.04. At $1.64\pi$, the fluctuations can be seen to increase towards the bottom, reaching a peak at 3 mm (about $2\Delta$) over 0.04. Figure 10 also shows that adding a small concentration profile did not have a significant impact on the momentum of the solution, which can be seen in how closely it follows the case without sediment. The results also compare favorably with the experimental work of Jensen et. al. [37] on oscillatory flows in an intermittently turbulent regime (See Figure 10, diamond symbols). Doubling the grid resolution in the simulation only had a small effect, such that the root-mean-squared velocity perturbations in the streamwise direction did not exceed 1% at the phase of maximum flow (see for example, Figure 10, first panel). This validates the ability of our solver to properly handle sediment concentrations within the bottom boundary layer under oscillating flow.

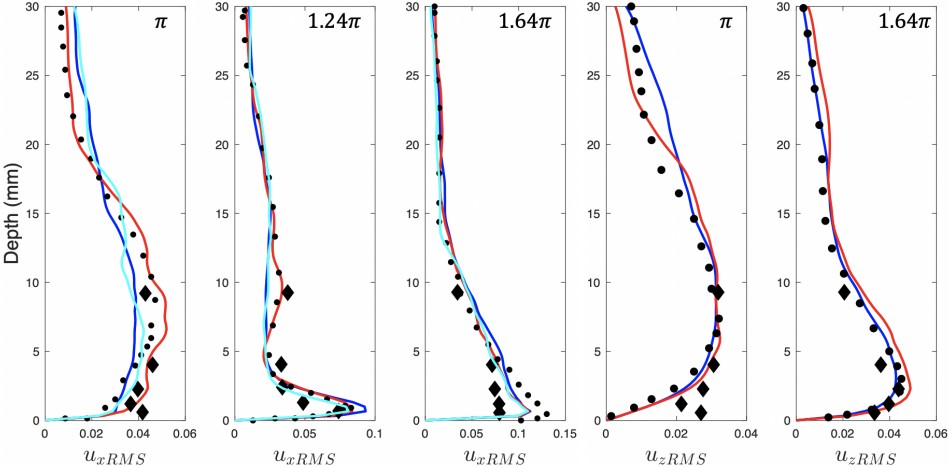

**Figure 10.** Phase-averaged normalized root mean squared velocity perturbations in the streamwise and vertical directions. Results are shown for the test case initialized with no concentration (blue), and the test case initialized with a concentration gradient (red). A comparison is made with with the numerical results of Costamagna et al. [36] (black dots), as well as the experimental results of Jensen et. al. [37] (black diamonds). The doubled resolution test case is denoted by the cyan line.

The volume fraction field is a tracer and its behavior through the domain provides insight into the development of turbulence over time. Figure 11 shows the volume fraction on an x–z plane cross-section through the domain. From this, the time evolution of turbulence can be seen, as the sediment goes from a well ordered initial gradient (Figure 11a,b) towards well-mixed homogenized conditions (Figure 11d).

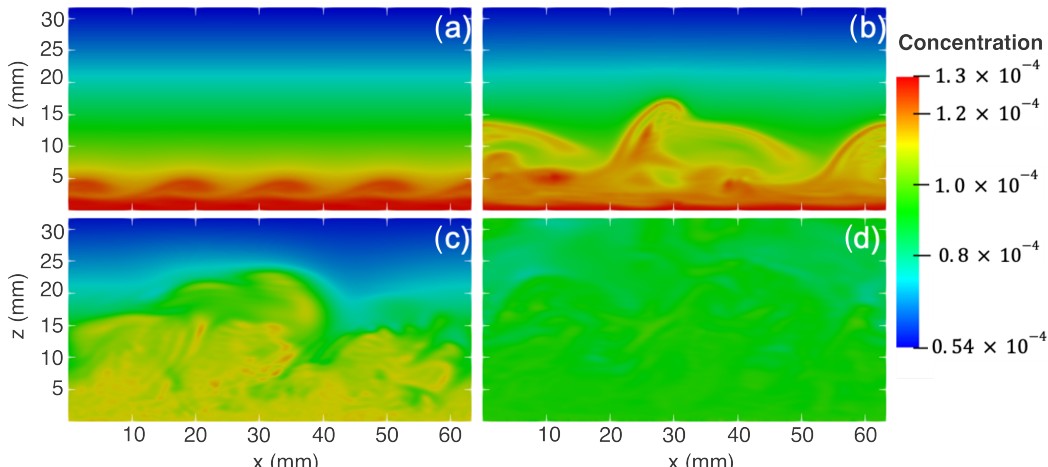

**Figure 11.** X–z cross-section through the center of the domain of concentration, taken through the first three cycles (**a**) 3 s, (**b**) 5 s, (**c**) 7 s, and (**d**) 13 s.

Figure 12 shows the growth and oscillations of the domain-averaged turbulent kinetic energy generated with the mudMixtureFoam solver through the first three flow cycles. Growth of turbulent kinetic energy occurs during the first cycle and quickly stabilizes for the second and third cycles. As expected, the small amount of density gradients in the test with the concentration profile did not alter significantly the final equilibrium value of the turbulent kinetic energy. These results demonstrate that the mudMixtureFoam solver can reproduce the generation of turbulence within the bottom boundary layer, as well as handle small concentrations of sediment, without having a substantial effect on the solution of the momentum.

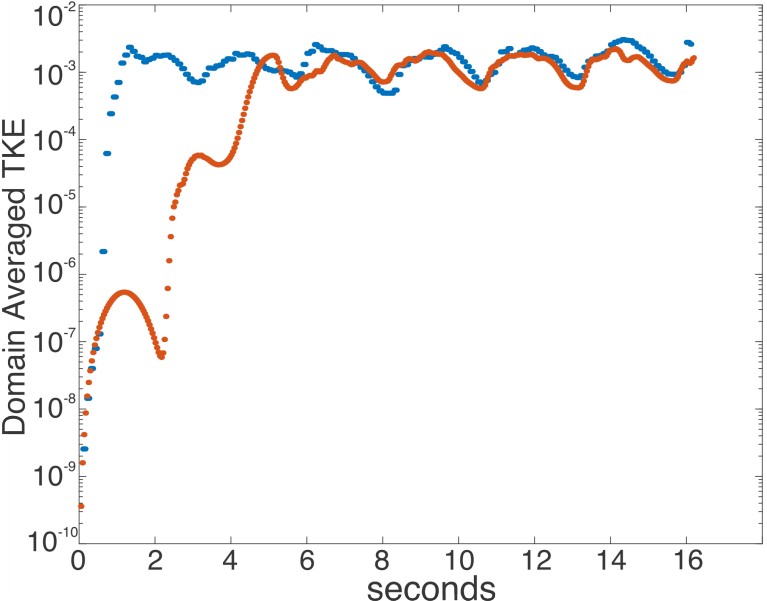

**Figure 12.** Log scale of turbulent kinetic energy over time for the test case initialized with no concentration (blue), and with a concentration gradient (red).

### 4.3. Two-Phase Flow in Oscillatory Boundary Layer

Here, we replicate the simulation of suspended fine sediments under an oscillatory flow from Ozdemir et al. [19]. In their methodology, the Boussinesq approximation was assumed for fluid flow, and the resulting momentum equation was used to formulate a model designed to capture the interactions of sediment properties and turbulence in the bottom boundary layer. Reproducing these numerical cases will give further validation to

the applicability of our non-Boussinesq model to fluid flows under such conditions, and facilitate the expansion of the modeled conditions to include denser suspensions.

An oscillating free-stream velocity of 0.56 m/s and a period of 10 s were selected to simulate highly energetic shelf conditions. This resulted in a Stokes boundary length of 1.8 mm and a Stokes–Reynolds number of 1000. The domain used in the previous section was extended vertically to achieve consistency with Ozdemir et al. This extension resulted in a domain with a vertical span of 60 Stokes boundary lengths. Length scales were normalized using this boundary layer thickness, and velocity scales were normalized using the maximum free stream velocity, $U_0$. Ozdemir et al. [19] made use of a pseudo-spectral method to resolve the initialization of turbulence at the bottom boundary, which we cannot reproduce with our second order spatial accuracy. Instead, we made use of the bottom waviness (Equation (17)) approach that was utilized by Costamagna et al. [36] and Mazzuoli et al. [40] to provide the infinitesimal "kick" that triggers the natural transition to turbulence.

The domain averaged sediment concentration was held to be 0.001 by volume within the domain, and initialized with the exponential profile $c = 7.04 \times V_s \times exp(-111 \times V_s \times z)$, where $c$ is the volumetric concentration of sediment, and $z$ is the non-dimensional vertical position vector. A bulk Richardson number of $1 \times 10^{-4}$, and Stokes–Reynolds number of 1000 were chosen for consistency with Ozdemir's numerical simulations.

We ran cases 2 and 3 from Table 3 with the mudMixtureFoam solver and compared the results to the digitized data output of Ozdemir et al. [19]. Points of comparison are the behavior of the along-stream velocity profiles and the profiles of concentration. Values are taken on three different phases of the oscillating cycle: $5\pi/6$, $\pi$, and $7\pi/6$. These phases correspond to the decelerating phase, the zero freestream velocity phase, and the negatively accelerating phase, respectively.

**Table 3.** Parameters in the two simulation cases from Ozdemir et al. [19].

| Case | $Re_\Delta$ | Ri | Vs | $U_0$ | T |
|------|------------|------|------|------|------|
| Case 2 | 1000 | $1 \times 10^{-4}$ | $4.5 \times 10^{-4}$ | 0.56 m/s | 10 s |
| Case 3 | 1000 | $1 \times 10^{-4}$ | $7.5 \times 10^{-4}$ | 0.56 m/s | 10 s |

The phase-averaged concentration profiles in our simulations (blue lines) are in good agreement with those of Ozdemir et al. (red dots) and formed a lutocline, a sharp concentration gradient, at approximately 25 Stokes boundary lengths from the bottom (Figure 13 top row). The lutocline separates the upper fluid layer (with low magnitudes of turbulence) from the lower turbulent layer. Likewise, our simulations reproduced the vertical behavior of turbulence in relation to the lutocline, where the region below the lutocline is characterized by substantially increased values of turbulent velocity fluctuations (Figure 13 bottom row). The presence of a distinct lutocline is a characteristic behavior of the intermittently turbulent flow regime. The flow in this layer is highly energetic, displaying high values of streamwise root mean squared velocity, as can be seen in the bottom row of Figure 13. We found a similar agreement between our model and Ozdemir et al. for case 3 in Table 3.

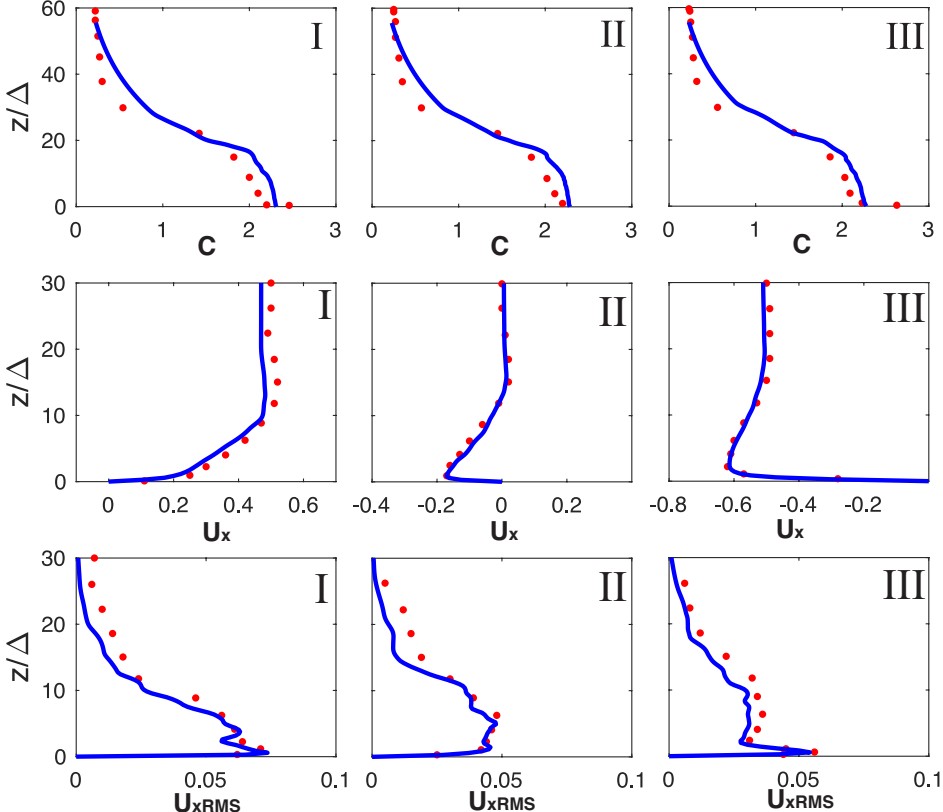

**Figure 13.** Phase-averaged profiles of the sediment concentration, normalized streamwise velocity, and RMS of the streamwise perturbations for mudMixtureFoam (Blue) are compared against the data from the Case 2 simulation of Ozdemir et al. [19] (red). Taken at three phases of the oscillating cycle: $5\pi/6$ (I), $\pi$ (II), and $7\pi/6$ (III).

## 5. Conclusions

In this paper, we presented a two-phase model designed to simulate the interaction of bottom turbulence with density stratification due to resuspended sediment. This was achieved by coupling a viscoplastic model to the non-Boussinesq momentum equations. The viscoplastic model approximates the behavior of a Bingham plastic using empirically determined dependencies of the dynamic viscosity and yield stress on sediment concentration. Sediment-specific parameterizations can be used to adjust the viscosity model towards different mud compositions, resulting in a broad applicability of the overall model to a diverse set of conditions. The solver was implemented in Openfoam by modifying the pressure solver algorithm to account for a non-solenoidal fluid velocity in a two-phase system.

The viscosity model component of the solver was successfully validated against steady flow of a Bingham plastic fluid over a sphere, accurately recreating the behavior of viscosity around the solid sphere. The model demonstrated good agreement with previous simulations, with a direct comparison of the viscosity profiles with Gavrilov's simulations [24] showing root mean square deviations not exceeding 1 percent near the sphere surface, with most deviations falling below 0.1 percent. The drag force over the surface of the sphere for flow conditions with a Bingham number of 1 and Reynolds numbers of both 0.1 and 100 converged to the values predicted by the drag force equation proposed by Gavrilov (Equation (11)).

MudMixtureFoam was able to produce intermittently turbulent flow within the bottom boundary layer under an oscillating wave. The accuracy of turbulence production was validated by comparison of the velocity perturbation profiles throughout all phases of flow

to a prior simulation carried out by Costamagna [36] and the laboratory experiments of Jensen [37]. The profiles demonstrated the model's ability to capture the behavior of the turbulent layer with peak streamwise perturbations reaching magnitudes over 10 percent of the freestream velocity and vertical perturbations reaching magnitudes up to 5 percent of the freestream velocity and occurring at distances $1 - 2\Delta$ from the bottom.

Finally, we carried out numerical simulations of intermittently turbulent sediment-laden flow and found good agreement with the prior simulation of [19] for a Stokes–Reynolds number of 1000 and a bulk Richardson number of $1 \times 10^{-4}$. Formation of a lutocline between 20 and 40 Stokes boundary layer depths was seen, and an increase in the settling velocity by 67 percent resulted in the location of the lutocline dropping by approximately 5 mm. Below this lutocline, the turbulent velocity perturbations displayed a marked increase in magnitude, with normalized velocity values exceeding 0.05 within the first 10 Stokes boundary lengths from the bottom. As the lutocline forms directly above the turbulent layer, this provides a measure of the thickness of the intermittently turbulent bottom boundary layer, which itself is sensitive to the settling velocity. Given these validations, we believe that MudMixtureFoam would be a useful tool to investigate the effects of high nearbed concentration on turbulence in oscillatory bottom boundary layers. The numerical model described in this work can be used to refine the parameterization of the effects of sediment driven density stratification in sediment diffusion models of silty suspensions [41].

**Author Contributions:** Conceptualization, I.A. and J.S.; Methodology, I.A. and J.S.; Software, I.A., J.S., S.B. and N.K.; Validation, I.A.; Writing—original draft, I.A.; Writing—review & editing, I.A. and J.S.; Visualization, I.A.; Supervision, J.S.; Project administration, J.S. All authors have read and agreed to the published version of the manuscript.

**Funding:** This research was performed while Ian Adams held a National Research Council Research Associateship award at the U.S. Naval Research Laboratory. Nathan Keane was supported through the Naval Research Enterprise Internship Program at the U.S. Naval Research Laboratory. Julian Simeonov and Samuel Bateman were supported through base funding of the U.S. Naval Research Laboratory. This work was supported in part by a grant of computer time from the DoD High Performance Computing Modernization Program.

**Data Availability Statement:** Simulation data is available by request to the authors

**Conflicts of Interest:** The authors declare no conflict of interest.

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
