# Peer review of "A Bingham Plastic Fluid Solver for Turbulent Flow of Dense Muddy Sediment Mixtures"

_fluids, doi:10.3390/fluids8060171_

Round 1

Reviewer 1 Report

The problem solved in the study is the development of a new numerical product for modeling the behavior of the Bingham fluid, the dynamics of cohesive sediments.

The topic original and relevant in the field. It address a specific gap in the field. For problems of this kind, it is necessary to have different models that allow you to explore different aspects of the problem.

Compared with other published material, the developed model uses other parametrizations for, for example, the rate of deposition of sand particles.

The conclusions consistent with the evidence and arguments presented and they address the main question posed.

About the references appropriate, there are not many specific works on the problem in the list, but there are simply not many of them. The list of links is quite representative.

Tables and figures are representative and understood.

Minor:

The problem of validation of numerical simulation results with the help of laboratory experiments is ignored. Comparison is performed only with other numerical results.

I suggest that the authors think about validating the numerical calculation. Try to find data from laboratory experiments that can be used for this purpose. 

Author Response

Response to Reviewer 1 Comments

The problem solved in the study is the development of a new numerical product for modeling the behavior of the Bingham fluid, the dynamics of cohesive sediments.

The topic original and relevant in the field. It address a specific gap in the field. For problems of this kind, it is necessary to have different models that allow you to explore different aspects of the problem.

Compared with other published material, the developed model uses other parametrizations for, for example, the rate of deposition of sand particles.

The conclusions consistent with the evidence and arguments presented and they address the main question posed.

About the references appropriate, there are not many specific works on the problem in the list, but there are simply not many of them. The list of links is quite representative.

Tables and figures are representative and understood

Author’s Response: Thank you for your comments.

  1. The problem of validation of numerical simulation results with the help of laboratory experiments is ignored. Comparison is performed only with other numerical results. I suggest that the authors think about validating the numerical calculation. Try to find data from laboratory experiments that can be used for this purpose. 

Author’s Response: This point is well taken. We have added a comparison to the experimental work of Jensen et. al. who examined the turbulence of the bottom boundary layer under oscillating flow and intermittently turbulent conditions on lines 369-371. The revised manuscript now includes the root-mean-squared of the streamwise and vertical velocity perturbations of the laboratory experiment in Figure 10.

Reviewer 2 Report

Nice job! Hope you can release the code upon the publication of this work. It will help the community.

None.

Author Response

Response to Reviewer 2 Comments

Nice job! Hope you can release the code upon the publication of this work. It will help the community.

Author’s Response: Thank you! This is our hope too.

Reviewer 3 Report

In this study, a Bingham plastic fluid solver for turbulent flow of dense muddy sediment mixtures in OpenFOAM was developed and validated with three cases. In all cases, the new solver produced excellent agreement with the previous results. This topic is interesting and the new solver can be useful for future sediment transport studies. I have some suggestions before the paper can be accepted, which are:

1.     Please update the recent references. There is only one reference after 2020.

2.     There is no sensitivity analysis for grid number. How do we know the grid number is sufficient?

3.     What do the color mean in Figures 4, 5, and 6?

4.     What is the y axis in Figure 7?

Quality of English Language is good.

Author Response

Response to Reviewer 3 Comments

This topic is interesting and the new solver can be useful for future sediment transport studies. I have some suggestions before the paper can be accepted, which are:

Author’s Response: Thank you for taking the time to review our paper and make suggestions.

  1. Please update the recent references. There is only one reference after 2020.

Author’s Response: Several newer references from 2022 and 2023 have been added in the introduction and conclusions to provide a more recent context for resuspension and transport of muddy sediment, and to frame future applications of our work. These are included on lines 38-40, 54-58, 62-65, and 469-471.

  1. There is no sensitivity analysis for grid number. How do we know the grid number is sufficient?

Author’s Response: A sensitivity test was performed by doubling the grid resolution. Results show that root-mean-squared perturbation profiles at the phase of maximum flow does not exceed ~1% when the resolution is increased. A discussion of this test and its implications was added to the description of the results on lines 371-373, and the increased resolution results were added to figure 10.

  1. What do the color mean in Figures 4, 5, and 6?

Author’s Response: The figure captions were altered to include the definition of the color contours denoting the nondimensionalized viscosity at integer values in the range of 1-10

  1. What is the y axis in Figure 7?

Author’s Response: The axis label has been included to show that this is the nondimensionalized viscosity magnitude